# Characterization and Transcriptome Analysis of Maize Small-Kernel Mutant *smk7a* in Different Development Stages

**DOI:** 10.3390/plants12020354

**Published:** 2023-01-12

**Authors:** Jing Wang, Hongwu Wang, Kun Li, Xiaogang Liu, Xiaoxiong Cao, Yuqiang Zhou, Changling Huang, Yunling Peng, Xiaojiao Hu

**Affiliations:** 1College of Agronomy, Gansu Agricultural University, Lanzhou 730070, China; 2National Engineering Research Center of Crop Molecular Breeding, Institute of Crop Sciences, Chinese Academy of Agricultural Sciences, Beijing 100081, China; 3Gansu Provincial Key Laboratory of Aridland Crop Science, Gansu Agricultural University, Lanzhou 730070, China

**Keywords:** maize, *smk7a* mutant, IAA, starch synthesis, transcriptome

## Abstract

The kernel serves as a storage organ for various nutrients and determines the yield and quality of maize. Understanding the mechanisms regulating kernel development is important for maize production. In this study, a small-kernel mutant *smk7a* of maize was characterized. Cytological observation suggested that the development of the endosperm and embryo was arrested in *smk7a* in the early development stage. Biochemical tests revealed that the starch, zein protein, and indole-3-acetic acid (IAA) contents were significantly lower in *smk7a* compared with wild-type (WT). Consistent with the defective development phenotype, transcriptome analysis of the kernels 12 and 20 days after pollination (DAP) revealed that the starch, zein, and auxin biosynthesis-related genes were dramatically downregulated in *smk7a*. Genetic mapping indicated that the mutant was controlled by a recessive gene located on chromosome 2. Our results suggest that disrupted nutrition accumulation and auxin synthesis cause the defective endosperm and embryo development of *smk7a*.

## 1. Introduction

Maize (*Zea mays*) is one of the most important crops in the world. The maize kernel is the storage organ that contains the essential components for plant growth and reproduction. The maize kernel consists of the embryo, endosperm, and surrounding pericarp. The embryo consists of the shoot apical meristem, root apical meristem, scutellum, and leaf primordium and has the potential to develop into a new plant [1]. The endosperm accounts for 85% of the kernel’s weight and stores about 80% of the starch and 8–15% of the protein. The development of the endosperm begins with the fertilized central cells, followed by cellularization and differentiation into four main types of cells: the basal endosperm transfer layer (BETL), the aleurone layer (AL), the starchy endosperm (SE), and the embryo-surrounding region (ESR) [2,3,4,5,6,7]. Maize kernel development is a complex and dynamic process and is regulated by a large number of genes. With the development of molecular techniques, many genes that are crucial for kernel development have been identified, and the regulatory mechanisms have also been uncovered [8].

Regarding embryo development, a number of embryo-specific (*emb*) genes have been studied, such as *LEM1*, *EMB8516*, *EMB14*, *EMB12*, and *EMB16* [9,10,11,12,13], all of which have been implicated in plastid function. These studies have demonstrated that plastid protein translation is essential to embryogenesis. Regarding endosperm development, the functional genes involved in the development of the major endosperm cell types have also been elucidated. The AL is a cuboidal cell layer that forms on the epidermal layer of maize endosperm cells. Several genes have been reported to control the fate of the AL cells via positional signaling, including *CR4* [14], *DEK1* [15,16], *SAL1* [17], and *THK1* [18,19]. The BETL, a unique endosperm cell layer, plays an important role in nutrient transport and signaling between the filial and maternal tissues [20]. There are two key genes that affect the nutrient uptake and partitioning in the BETL region. The first gene is *MN1*, which is located in the BETL and the pedicel (PED) and catalyzes the conversion of sucrose into glucose and fructose [21,22]. The second gene *ZmSWEET4c* encodes a sugar transporter [23], and it can transport hexose to the starchy endosperm and embryo for its development [24,25]. ZmMRP-1 is a well-known transcriptional regulator in BETL differentiation. It controls the expression of BETL-specific genes, including *ZmTCRR-1* and *ZmTCRR-2* [26,27], *MEG1* [10], and *BETL-1*, *BETL-2*, *BETL-9*, and *BETL-10*. The SE is the major storage tissue for starch synthesis and protein storage in the endosperm. Starch biosynthesis requires multiple classes of enzymes, including adenosine diphosphate (ADP)-glucose pyrophosphorylase (AGPase), starch synthases (SSs), starch branching enzymes (SBEs), and starch debranching enzymes (DBEs) [28]. The key genes have been identified in mutant studies, such as *SH2*, *BT2*, *DU1*, and *SU1* [29,30]. The loss of the function of these genes results in disturbed starch synthesis and defective kernel development. The ESR is formed from small dense cytoplasmic cells surrounding the young embryo. The functions of the ESR are providing nutrition to the embryo, defense from pathogens, and signaling at the embryo-endosperm interface. The *ESR1*, *ESR2*, and *ESR3* genes have been suggested to play an important role in regulating embryo–endosperm signaling [31].

Although significant progress has been made in studies on maize kernel development, additional knowledge is still needed to reveal the complex interaction network during embryo and endosperm development. In this study, a thorough analysis of a small kernel mutant *smk7a* was conducted using cytology, biochemistry, and transcriptomics. These results provide a foundation for further gene cloning and elucidation of the mechanism involved in maize kernel development.

## 2. Results

### 2.1. smk7a Mutant Exhibited Defective Embryo and Endosperm Development

Compared with WT kernels, *smk7a* exhibited defective kernel development and a lower 100-kernel weight (Figure 1A–C and Appendix A). To observe the structural and cytological changes, we carried out paraffin sectioning on WT and *smk7a* kernels collected at 12, 16, 20, 25, and 30 DAP. Longitudinal sections of the whole seed revealed that *smk7a* showed a small and shrunken kernel phenotype, with severely arrested embryogenesis and endosperm development. The WT embryo developed to the late embryogenesis stage, with established scutellum, coleoptiles, and shoot apical meristem (SAM) at 12 DAP; differentiated primary leaves at 16 DAP; established root apical meristem (RAM) at 20 DAP; and good differentiation at 30 DAP. However, the development of the *smk7a* embryos was severely delayed at 12 DAP, and it did not reach the coleoptilar stage with differentiated scutellum, coleoptile, SAM, and RAM until 30 DAP (Figure 1D). The WT endosperm was well developed and starch-filled from 12 to 30 DAP, while the endosperm of *smk7a* was small and underdeveloped, and a gap was observed between the endosperm and epidermis (Figure 1D). Further observations revealed that the transfer cells in the BETL of the WT were large and exhibited clear cell wall ingrowths at 12 DAP, and the flange wall ingrowth became more dense at 20 DAP. In contrast, the BETL cells in the *smk7a* kernels were mostly small and lacked extensive cell wall ingrowth (Figure 1E).

We also found that *smk7a* had a 56.3% reduction in the endosperm cell number compared to the WT, and the endosperm cell size was 188.7% larger than that of the WT at 20 DAP (Figure 2A–C and Table 1). Although the endosperm cell size was larger, the starch granules of *smk7a* appeared to be less filled than the endosperm of the WT (Figure 2A). Scanning electron microscopy (SEM) was used to investigate the starch accumulation in the endosperm at 34 DAP. The results revealed that the endosperm cells of the WT were filled with starch granules, while there were few granules in the endosperm cells of *smk7a*, and irregular polygonal granules were observed (Figure 2E,F).

### 2.2. smk7a Mutant Had Lower Starch, Storage Protein, and Free IAA Contents

To further determine the underlying biochemical basis for the defective phenotype of the *smk7a* mutant, we measured the soluble sugar, total starch, and protein contents of the WT and *smk7a* kernels. Based on our results, the content of the soluble sugars (including glucose, fructose, and sucrose) of the *smk7a* kernels was 1.40–2.70-fold higher than that of the WT kernels at 12 DAP, and it was 3.72–4.07-fold higher at 20 DAP (Figure 3A–C). In contrast, the starch content was significantly lower, i.e., 4.25-fold and 4.67-fold at 12 and 20 DAP, respectively (Figure 3D). In addition, the starch content of *smk7a* was only 87.8% that of the WT in the mature stage (Figure 3D). We also noticed that the total soluble sugar content of the WT decreased by 40.3% from 12 DAP to 20 DAP, while the starch content increased, which represents the balanced conversion of sugar to starch. However, for *smk7a*, when the starch content increased, the soluble sugar content also increased by 22.3% from 12 to 20 DAP. These results suggest that the sugar metabolism and starch synthesis were disturbed in the *smk7a* kernels in the filling stage. Quantitative analysis of the protein contents revealed that the zein protein was also significantly lower in *smk7a* and was 71.7%, 76.7%, and 94.0% that of the WT at 12 DAP, 20 DAP, and in the mature stage, respectively (Figure 3E).

IAA has been reported to play an important role in kernel development. We compared the free IAA contents of the WT and *smk7a* at 12 and 20 DAP. Based on our UHPLC-ESI-MS/MS analyses, the free IAA content of *smk7a* was 58.3% and 83.2% that of the WT at 12 DAP and 20 DAP, respectively (Figure 3F). These results suggest that IAA synthesis was suppressed in *smk7a* in the early development stages.

### 2.3. Transcriptome Comparison of Developing Kernels from WT and smk7a

To reveal the gene expression changes between the WT and the *smk7a* mutant, we conducted RNA sequencing (RNA-seq) using the kernels collected at 12 and 20 DAP. On average, 48.2 million clean reads were obtained for each sample, and 89.2–94.9% of the reads were uniquely mapped to the maize B73 reference genome (Appendix A). With a threshold fold change of >2 and an adjusted *p* value of <0.05, a total of 3405 and 2747 differentially expressed genes (DEGs) were identified between *smk7a* and WT for the 12 and 20 DAP kernels, respectively. Further classification of the DEGs revealed that there were 1522 downregulated genes and 1883 upregulated genes at 12 DAP, and there were 1124 downregulated genes and 1623 upregulated genes at 20 DAP (Appendix A). Since the BETL development was impaired in *smk7a*, we checked the expression levels of the BETL-specific genes. At 12 DAP, six genes were significantly downregulated in *smk7a*, namely, *BETL3*, *BETL4*, *BETL9*, *TCRR1*, *TCRR2*, and *SWEET4C*, while only the *SWEET4C* gene was significantly downregulated at 20 DAP. The expression levels of *BETL4*, *BETL9*, *TCRR1*, and *SWEET4C* in *smk7a* and WT were further verified using the quantitative real-time polymerase chain reaction (qRT-PCR) method (Appendix A). The reduced expression of the BETL-specific genes in *smk7a* validated the development defects of the BETL cells.

To associate functions with the DEGs, we performed gene ontology (GO) enrichment analysis. The 20 most significantly (*p* < 0.05) enriched GO terms in the biological process, molecular function, and cellular component categories are presented in Appendix A. At 12 DAP, the downregulated genes were mainly enriched in the BP term “starch biosynthetic process”, the MF term “nutrient reservoir activity”, and the CC terms “chloroplast and amyloplast”, while the upregulated genes were mainly enriched in the BP terms “cell differentiation”, “lipid catabolic process”, and “hydrogen peroxide catabolic processes”, the MF term “heme binding”, and the CC term “extracellular region” (Appendix A). At 20 DAP, the downregulated genes were significantly enriched in the BP term “secondary metabolite biosynthetic process”, the MF term “nutrient reservoir activity”, and the CC term “extracellular region”, while the upregulated genes were mainly enriched in the BP terms “defense response” and “hydrogen peroxide catabolic processes”, the MF term “sequence-specific DNA binding”, and the CC terms “extracellular region” and “plant type cell wall” (Appendix A).

We further conducted Kyoto Encyclopedia of Genes and Genomes (KEGG) enrichment analysis of the DEGs. The results revealed that the “starch and sucrose metabolism” and “plant hormone signal transduction” were the most significantly enriched pathways for the downregulated genes in both development stages (Figure 4A,B), while the “fatty acid biosynthesis” and “glycolysis/gluconeogenesis pathways” were only enriched at 12 DAP (Figure 4A). For the upregulated genes, the “phenylpropanoid biosynthesis” pathway was significantly enriched in the two development stages (Figure 4A,B).

These results suggest that the expression of the genes with starch synthesis and nutrient accumulation functions was suppressed in *smk7a* compared to that in the WT in both development stages. The suppressed transcriptional activity of these genes could be the major reason for the defective development of the *smk7a* endosperm and embryo.

### 2.4. Suppressed Expression of Genes Involved in Starch Synthesis and Protein Storage in smk7a

Starch and storage protein accumulation is essential for kernel development and yield formation. Our RNA-seq results revealed that the genes that participated in the “starch and sucrose metabolism” pathway were significantly downregulated in *smk7a* compared to that in the WT (Figure 4). In the starch synthesis process, sucrose, glucose, and fructose are first imported into the endosperm cell and are metabolized to hexose phosphates. Then, these hexose phosphates serve as substrates for starch biosynthesis in the amyloplast [32,33]. The sucrose synthase *SH1*, *SUS2,* and *SUS3* functions for converting sucrose into fructose and uridine-diphosphoglucose (UDPG) were downregulated in the *smk7a* by 6.9–13.9-fold at 12 DAP and 0.9–1.9-fold at 20 DAP. The hexokinase (HEX) phosphorylates hexoses forming hexose phosphate, *HEX1*, and *HEX5* genes were downregulated in *smk7a* by 2.3–5.0-fold and 1.0–1.8-fold at 12 and 20 DAP, respectively (Figure 5A). The interconversions among fructose 6-phosphate (F6P), glucose 6-phosphate (G6P), glucose 1-phosphate (G1P), and UDPG were catalyzed by phosphohexose isomerase (PHI), phosphoglucomutase (PGM), and UDP-glucose pyrophosphorylase (UGP), and four related genes were also found to have a reduced expression in *smk7a* (Figure 5A). The ADP-glucose pyrophosphorylase genes (APGs) catalyze the first step in starch biosynthesis, which is also the limiting step. Five APGs were significantly downregulated in *smk7a*, including *SH2 and BT2*, the expressions of which were decreased by 9.2- and 26-fold, respectively, at 12 DAP (Figure 5A). During the starch synthesis in the amyloplast, the expressions of the genes with amylose synthase (such as *WX1*, *GBSSI*, and *GBSSIb*), amylopectin synthase (*SSIV*, *SSV*, and *SSI*), branching enzyme *(SBE)*, and starch debranching enzyme (*SU1*) functions were all significantly reduced (Figure 5A and Appendix A). To further verify these results, the expression levels of *SH1*, *SH2*, *WX1*, *BT2*, *AGP*, and *SSI* in *smk7a* and the WT were examined using the qRT-PCR method (Appendix A). The overall downregulation of the expression of the genes with starch and sucrose metabolism pathway functions was in agreement with the lower starch content in *smk7a* at 12 and 20 DAP and in the mature stage.

We further checked the expression of the sugar transporter genes since the soluble sugar content was higher in *smk7a* compared to that in the WT. We detected 19 transporters, including SWEETs, sugar transporter proteins (STPs), and sucrose transporters (SUTs), which were expressed differently in *smk7a* and the WT, and 14 of these genes were upregulated in at least one development stage (Figure 5B and Appendix A). These results suggest that suppressed starch synthesis may act as a feedback signal for stimulating sugar transport from the maternal tissue to the kernel endosperm, even when the development of the BETL cells is defective.

Moreover, we also found that 23 genes related to protein storage under the GO term “nutrient reservoir activity” (GO: 0045735) were significantly downregulated in *smk7a*. It should be noted that central player transcript factors O2, O11, and PBF were also downregulated in *smk7a* in both development stages (Figure 5C and Appendix A). These results provide a sufficient explanation for the reduced zein protein content in *smk7a* compared to that in the WT.

### 2.5. Disrupted Expression of Genes Related to IAA Synthesis and Signaling in smk7a

Studies have reported that phytohormones such as auxin, cytokinins, and brassinosteroids play an important role in plant seed development. In this study, the IAA content in *smk7a* was significantly lower, and our RNA-seq results also revealed that the expression of the key genes involved in auxin synthesis was decreased in *smk7a* (Figure 6 and Appendix A). The tryptophan aminotransferase (*TAA1/TAR*) genes convert tryptophan to indole-3-pyruvic acid (IPA). Mutations in the *TAA1/TAR* of *Arabidopsis* lead to a dramatic reduction in the free IAA level. Three TAR genes were found to be downregulated in *smk7a* at 12 DAP, while the expression level of *TAR1* and *TAR3* increased at 20 DAP. *ZmYUC1*, a flavin-monoxygenase, which is the key enzyme related to auxin biosynthesis in maize endosperm, was significantly downregulated (by 6.1-fold) in *smk7a* at 12 DAP. The suppressed IAA synthesis also caused a disturbance of the IAA signaling. The expression levels of the core auxin signaling factors, including 10 auxin/indole-3-acetc acid (Aux/IAA), five auxin responsive factors (ARFs), six Gretchen Hagen3 (GH3), and seven small auxin RNA (SAUR) were changed in *smk7a* (Figure 6). In the presence of low-level intracellular auxin, AUX/IAA proteins inhibit the function of ARF proteins. Based on our RNA-seq, nine of the 10 AUX/IAAs were upregulated in *smk7a*, while all five ARFs were downregulated in *smk7a* (Figure 6). These results suggest that auxin synthesis and signaling were suppressed in *smk7a*, which may be the major cause of the reduced number of cells in the *smk7a* endosperm.

### 2.6. Genetic Analysis and Fine Mapping of the smk7a Mutant

Our previous study reported that *smk7a* mutant was controlled by a recessive gene located in the 120 kb region on chromosome 2 [34]. To further narrow down its location, we genotype another F_2_ population of 1056 homozygous mutant kernels using newly developed insertion and deletion markers (Appendix A); however, no new recombination was detected at this interval. The RNA-seq results revealed that only three genes exhibited an expression level greater than 1 fragment per kilobase of the transcript per million fragments (FPKM) in at least one of the samples in this region. *Zm00001d001819* encodes an N-acetylglucosaminyl-phosphatidylinositol de-N-acetylase, *Zm00001d001824* encodes a Dof zinc finger protein, and *Zm00001d001825* encodes a transducin/WD40 repeat-like superfamily protein.

## 3. Discussion

The maize kernel has been used extensively as a valuable model for cereal kernel development studies. The development of the maize kernel can be divided into three stages: the early development, filling, and dehydration stages [1]. In the early development stage (about 0–15 DAP), the embryo and endosperm cells are quickly divided and differentiated into specific tissues. In the filling stage (about 15–45 DAP), the embryo and endosperm start to accumulate starch and to store protein lipids and vitamins [35]. Then, the kernel enters the mature stage and begins to dehydrate (about 45–60 DAP) [36]. Kernel mutants with developmental defects in different stages have been characterized, and a great number of genes have been cloned [7,8]. These studies have provided valuable information for unveiling the gene regulation network of maize kernel development.

In this study, we characterized a small-kernel mutant *smk7a*, which exhibited arrested embryo and endosperm development compared to the WT. The small and pale *smk7a* kernels with irregular embryo and endosperm can be clearly distinguished from the WT at 12 DAP. These results suggest that the mutant phenotype of *smk7a* first occurred before 12 DAP. This phenotype of *smk7a* is quite similar to some kernel mutants with variant genes related to early maize kernel development. For example, the *dek15* mutant exhibits a defective endosperm and embryo in the early stage, and the embryos appear to be more severely affected than the endosperm. The *DEK15* encodes cohesin-loading complex subunit SCC4, and the loss of the SCC4 function leads to defects in the cell division and early kernel development [37]. Both embryo development and endosperm development are delayed in the *smk4-1* mutant compared with that in the WT at 8 DAP. *SMK4* encodes an E-subclass PPR protein, which is exclusively localized in mitochondria [38]. The *smk1* mutation leads to defects in the assembly of complex I, which results in the arrest of the embryo and endosperm in the early development stage [39]. Thus, we speculate that the functional gene controlling the *smk7a* phenotype may play an important role in early kernel development.

Our cytological observations also revealed that the BETL morphology is impaired in *smk7a*. The *smk7a* BETL developed less-elongated cells and had very few cell wall ingrowths. The RNA-seq results revealed that the expression levels of several BETL-specific genes were significantly downregulated. Defective BETL development usually results in reduced sucrose transport from the maternal plant to the endosperm, and it ultimately leads to suppressed starch synthesis and endosperm development [6,40,41]. For example, *ENB1* encodes a cellulose synthase 5 that directs the synthesis of cell wall ingrowths in maize BETL cells. The *enb1* mutant exhibits a drastic reduction in the formation of flange cell wall ingrowths in the BETL cells and impaired nutrient uptake and starch synthesis [22]. *ZmCTLP1* participates in lipid homeostasis in transfer cells. The loss of the function of *ZmCTLP1* results in irregular-shaped BETL cells with few wall ingrowths and significantly reduced starch, sugar, and protein contents in the mutant [4]. Based on our biochemical results, the starch content of *smk7a* was significantly lower, while the soluble sugar content was higher in *smk7a* compared to WT at 12 and 20 DAP. Further analysis revealed that the starch accumulation in the WT increased from 12 to 20 DAP, which was accompanied by a reduction of the soluble sugar content, indicating balanced conversion from soluble sugar to starch. In contrast, the soluble sugar content in *smk7a* increased from 12 DAP to 20 DAP. These results suggest that the sugar and starch metabolism was disrupted in *smk7a*.

The RNA-seq data revealed that at 12 and 20 DAP, the DEGs enriched in the “starch and sucrose metabolism” pathway were significantly downregulated. Starch began accumulating in the endosperm of the maize kernels at approximately 10 DAP, starting with the transport of sucrose (Suc) from the maternal plant to the endosperm. Suc was then converted into fructose (Fru) and glucose (Glu) by neutral invertase (NINV) or into Fru and UDPG by sucrose synthase [42]. The resulting Glu and Fru were then phosphorylated to generate hexose phosphates, including G6P, F6P, and G1P. G1P was then further activated by AGPase to produce ADP-glucose [43], which was the substrate for starch synthesis [44,45]. In our RNA-seq results, almost all of the DEGs involved in this sucrose metabolism process exhibited a decreased expression level in *smk7a*, including the well-known genes *SH1*, *SH2*, and *BT2*. *SH1* encodes sucrose synthase, and the loss of the function of *SH1* caused a significant reduction in the carbohydrates that flow to starch synthesis, ultimately contributing to the defects in the starch granule development and reduction of the starch content [46]. *SH2* and *BT2* encode AGPase and are rate-limiting enzymes in starch biosynthesis. Mutations in *SH2* and *BT2* resulted in greatly reduced starch production and increased sugar in the endosperm [47,48,49]. In *smk7a*, at 12 DAP, the expressions of *SH2* and *BT2* were decreased by 9.2- and 26-fold, respectively, compared to that in the WT. These results explain the increased soluble sugar contents of the *smk7a* kernels compared to that of the WT kernels. After the sugar was metabolized into ADP-glucose, starch was synthesized by multiple subunits or isoforms of five classes of enzymes, namely, AGP, SS, GBSS, SBE, and DBE [33]. In our RNA-seq, the expressions of amylose synthases WX1, GBSSI, and GBSSIb; amylopectin synthases SSIV, SSV, and SSI; branching enzyme SBE; and starch debranching enzyme SU1 were all significantly reduced in *smk7a* compared to those in the WT. All of the above results indicate that the process of starch synthesis is severely suppressed in *smk7a*, which could be the major reason for the defective phenotype of the *smk7a* kernel.

Our results also revealed that the accumulation of the major storage protein zein was significantly reduced in *smk7a* at 12 and 20 DAP and in the mature stage. Consistent with this, all of the zein genes in the “nutrient reservoir activity” term (GO: 0045735) were downregulated in *smk7a*. It should be noted that several key regulator factors, including O2, O11, and PBF, were also found to be downregulated in *smk7a*. O2 was the first bZIP transcription factor known to regulate the expression of zein genes by recognizing the O2 box in their promoters. Loss of the function of O2 results in a drastically decreased zein content [50]. Several studies subsequently reported that O2 not only acts as a master regulator of the zein-coding genes but can also physically interact with the prolamin-box binding factor (PBF) to regulate starch accumulation [51]. O11 is a central hub of the regulatory network for maize endosperm development and nutrient metabolism [52]. It works upstream to regulate carbohydrate metabolic enzymes and the key regulators (such as O2 and PBF). The decreased expression of these genes suggests that the mutation in *smk7a* has a noticeable effect on starch and nutrition accumulation. The candidate gene of *smk7a* may interact with these key regulatory factors and could play an important role in the complex regulatory network of maize kernel development.

IAA is the most abundant endogenous auxin in the maize kernel, and it plays an essential role throughout the entire kernel development process [53]. In *smk7a*, the free IAA content was only 58.3% and 85.0% that of the WT at 12 DAP and 20 DAP, respectively. These results suggest that IAA synthesis is suppressed in *smk7a*. The RNA-seq results revealed that the essential genes that participate in the IPA pathway for IAA biosynthesis, including *ZmYUC1* [54], *TAR1*, *TAR2*, and *TAR3* [55,56], are significantly downregulated in *smk7a*. Defective kernel (*dek*) mutants in maize have been shown to have altered levels of IAA. For example, *DE18* encodes the auxin biosynthesis gene *ZmYUC1*, and *de18* mutation causes a large reduction of the free IAA level compared with the WT, leading to a 40% reduction in the dry mass of the endosperm. Several studies have also reported that developing kernels may regulate growth by influencing the auxin level in response to the sugar concentrations. The *MN1* gene encodes a cell wall invertase [20,57], which is localized in the BETL and catalyzes the conversion of sucrose into glucose and fructose. Loss of the function of *MN1* results in pleiotropic changes, including reduced cell number and cell size, a lower kernel mass, and a lower free IAA level [58]. In addition, the *mn1* mutant also exhibits retarded wall ingrowth formation in the BETL cells. Gene expression analysis suggests that the expression of the *ZmYUC1* and *TAR1* genes is greatly reduced in *mn1* kernels. In *smk7a*, the BETL cells were impaired and wall ingrowth was reduced, which could be the result of disrupted sugar hormone signaling in the endosperm.

## 4. Materials and Methods

### 4.1. Plant Materials

In our previous study, a maize small-kernel mutant *smk7* was isolated from the EMS mutant library produced in our laboratory on the B73 genetic background [34]. However, its name is repeated with another kernel mutant reported in a recent study [59], and therefore, we renamed it as *smk7a*. Since the homozygote *smk7a* cannot grow into a normal plant, the mutants were maintained in heterozygote by continuous self-crossing and phenotype selection. The developing kernels of WT and *smk7a* used for cytological, biochemical, and transcriptome analysis in this study were collected from the same segregating ears of the M8 generation. The heterozygous plant (*smk7a*/WT) was crossed with Mo17 and Chang7-2 and then self-pollinated to generate F_2_ populations. The segregating F_2_ ear were used for genetic mapping of *smk7a*. Maize plants were grown under natural conditions at the Changping experimental station of the Institute of Crop Sciences, Chinese Academy of Agricultural Sciences, Beijing.

### 4.2. Light Microscopy and Scanning Electron Microscopy Observations of smk7a and WT Kernels

WT and *smk7a* kernels were acquired from the same heterozygous ear at 12, 16, 20, 25, and 30 DAP. Then, the kernels were cut along the longitudinal axis into half-sections and fixed at room temperature in formalin-aceto-alcohol (FAA) solution (90 mL of 70% ethanol, 5 mL of glacial acetic acid, and 5 mL of 37% formaldehyde). The fixed material was dehydrated in an ethanol gradient series (50%, 70%, 85%, 90%, and 100% ethanol), embedded in paraffin, and subsequently cut into 12 μm sections. The sections were stained with toluidine blue and observed under a microscope (BX53; Olympus, Tokyo, Japan).

The WT and *smk7a* kernels collected at 34 DAP were fixed in FAA. The fixed material was dehydrated in an ethanol gradient series and then treated with isoamyl acetate for 15 min twice to replace the remaining ethanol. Then, the material was subjected to critical point drying. The samples were then coated with Pt particles and analyzed under a scanning electron microscope (SU8020, Hitachi, Tokyo, Japan).

### 4.3. Measurement of Protein and Starch Contents

Mature WT and *smk7a* kernels were obtained from the same segregating ears at 12 and 20 DAP. After removing the pericarp and embryo, the endosperms of the WT and *smk7a* were each pooled and ground into powder. Three biological replicates were prepared for the starch and protein measurements. For the starch quantification, 100 mg of powder per sample was analyzed using an a-amylase/amyloglucosidase starch assay kit (K-TSTA, Megazyme, Ireland) according to the manufacturer’s instructions. The starch content of the mature seeds was determined as has been previously described by [60]. For the protein quantification, 50 mg of powder was used for the protein extraction according to the method described by [61]. Then, the protein content was measured using a Bradford Protein Assay Kit (PC0010, Solarbio, Beijing, China).

### 4.4. Measurement of Soluble Sugar and Free IAA Contents

The kernels of WT and *smk7a* collected at 12 and 20 DAP from the same heterozygous ear were ground into power in the presence of liquid nitrogen. The soluble sugar and IAA contents were determined by Wuhan Greensword Creation Technology Co., Ltd. (Wuhan, China) according to previously reported methods [62,63]. The soluble sugars were analyzed via ultra-high performance liquid chromatography-electron spray ionization-tandem mass spectrometry (UHPLC-ESI-MS/MS, Thermo Fisher Scientific, Inc., Waltham, MA, USA). The IAA analysis was performed using a Thermo Scientific Ultimate 3000 UHPLC system equipped with a TSQ Quantiva-Stage Quadrupole Mass Spectrometer (Thermo Fisher Scientific, Sunnyvale, CA, USA). Three biological replicates were prepared for each measurement.

### 4.5. RNA Sequencing Analysis

The total ribonucleic acid (RNA) was isolated from the kernels of WT and *smk7a* collected at 12 and 20 DAP using a RNA Easy Fast Extraction Kit (DP452, Tiangen, Beijing, China). A total of 1 μg of RNA per sample was used to construct sequencing libraries using a TruSeq Stranded mRNA LTSample Prep Kit (Illumina, San Diego, CA, USA) according to the manufacturer’s instructions, and index codes were added to attribute the sequences to each sample. Then, these libraries were sequenced using the Illumina HiSeqTM 2500 platform to generate 125 bp paired-end reads. Following the removal of low-quality reads and adaptor contaminants, the clean reads were mapped to the maize reference genome RefGen_V4 (ftp.ensemblgenomes.org/pub/plants/release-32/gff3/zea_mays/, accessed on 1 November 2018) using HISAT2 [64]. Only perfectly matching reads were further analyzed. The gene expression levels were estimated using the fragments per kilobase of the transcript per million fragments (FPKM) mapped. Differential expression analysis of the two groups was performed using DESeq2 [65]. The resulting *p* values were adjusted using Benjamini and Hochberg’s approach for controlling the false discovery rate. The genes with an adjusted *p*-value of <0.05 and a fold change of >2 or <0.5 were assigned as differentially expressed.

Gene ontology (GO) enrichment analysis and Kyoto Encyclopedia of Genes and Genomes (KEGG) enrichment analysis were conducted using R based on the hypergeometric distribution, and those with a q value of <0.05 were defined as significantly enriched terms or pathways.

### 4.6. RNA Extraction and Quantitative RT-PCR Verification

The quantitative real-time polymerase chain reaction (qRT-PCR) technique was used to verify the expression level of the differentially expressed genes (DEGs) related to the BETL development and starch synthesis process. The total RNA of the kernels from the WT and *smk7a* collected at 12 and 20 DAP was extracted using an RNA Easy Fast Extraction Kit (DP452, Tiangen, Beijing, China) and then transcribed into complementary deoxyribonucleic acid (cDNA) using qPCR RT Master Mix with gDNA Remover (KR116, TIANGEN, Beijing, China). The qRT-PCR reactions were carried out using Taq Pro Universal SYBR qPCR Master Mix (Q712-02, Vazyme, Nanjing, China). The qRT-PCR analysis was performed using a Bio-Rad CFX96 qRT-PCR system. The reaction was carried out using three biological replicates with three technical replicates, with tubulin as the endogenous control (*Zm00001d013367*). The relative expression values were calculated using the 2^−ΔΔCt^ method [66], and the *t*-test was conducted to analyze the significant differences. The specific primers for the qRT-PCR are listed in Appendix A.

## 5. Conclusions

In summary, we characterized a small-kernel mutant *smk7a* of maize. Our results revealed that: (1) The early development of the embryo and endosperm of *smk7a* is seriously affected. (2) The starch and zein protein accumulation is lower in *smk7a* compared to in WT, and the expression levels of the DEGs involved in these processes are significantly suppressed in *smk7a*. (3) The auxin (IAA) content is lower in *smk7a* compared to in WT, and the DEGs that participate in IAA synthesis and signaling are downregulated in *smk7a*. (4) *smk7a* is a recessive gene located at chromosome 2. We speculate that *smk7a* may be related to the sugar and hormone signaling in the maize kernel and may play a crucial role in embryo and endosperm development.

## Figures and Tables

**Figure 1 plants-12-00354-f001:**
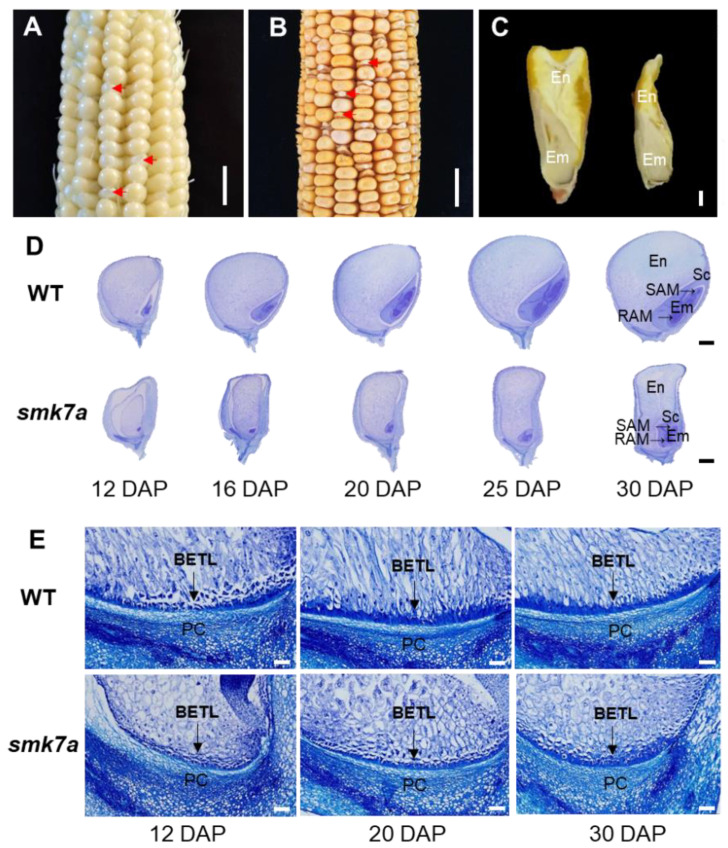
Phenotype features of the maize *smk7a* mutant. (**A**) An F_2_ ear of *smk7a*/WT at 12 DAP. The arrows indicate three randomly selected mutant kernels. The scale bar is 1 cm. (**B**) A mature F_2_ ear of *smk7a*/WT. The arrows indicate three randomly selected mutant kernels. The scale bar is 1 cm. (**C**) Examples of mature WT and *smk7a* kernels from the segregated F_2_. The scale bar is 1 mm. (**D**) Light microscopy analysis of immature kernels from the WT and *smk7a* collected at 12, 16, 20, 25, and 30 DAP. The scale bar is 1 mm. En, endosperm; Em, embryo; SAM, shoot apical meristem; RAM, root apical meristem. (**E**) Light microscopy analysis of BETL cells of the WT and *smk7a* kernels collected at 12, 20, and 30 DAP. DAP, days after pollination. BETL, Basal endosperm transfer layer; PC, placenta. The scale bar is 200 μm.

**Figure 2 plants-12-00354-f002:**
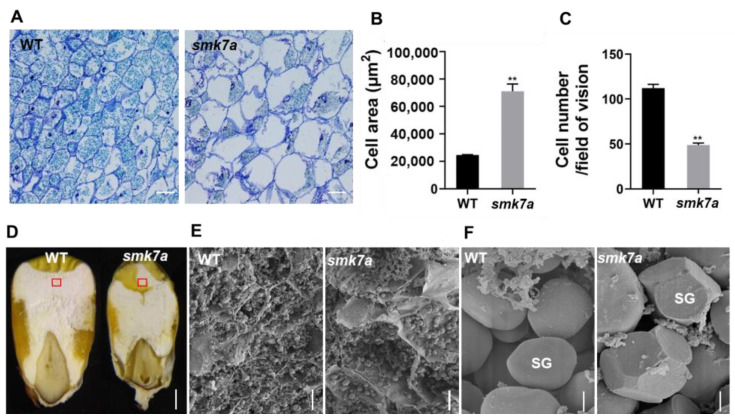
Comparison of the endosperm cells and starch granules in the WT and *smk7a*. (**A**) Light microscopy observation of the endosperm cells of the WT and *smk7a* kernels collected at 20 DAP. The scale bar is 100 μm. Comparison of the endosperm (**B**) cell area and (**C**) cell number between the WT and *smk7a*. The values are the means of the three replicates ± SD. Thirty cells were selected for statistical analysis for each replicate. ** denotes statistical significance at *p* < 0.01 according to the *t*-test. (**D**) Longitudinal section of the kernels collected from the WT and *smk7a* at 34 DAP. The red square indicates the region that was observed via scanning electron microscopy (SEM). The scale bar is 1 mm. (**E**) SEM analysis of the peripheral regions of the WT and *smk7a* endosperm collected at 34 DAP. The scale bar is 100 μm. (**F**) SEM analysis of the starch granules of the WT and *smk7a* endosperm. SG, starch granule. The scale bar is 10 μm.

**Figure 3 plants-12-00354-f003:**
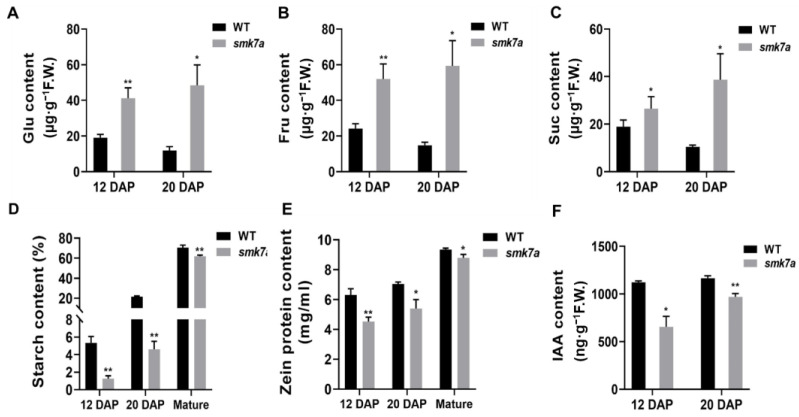
Comparison of biochemical values of kernels from the WT and *smk7a* mutant. (**A**–**C**) Comparison of the soluble sugar contents of the WT and *smk7a* at 12 and 20 DAP. Glc, glucose, Fru, fructose, Suc, sucrose. (**D**) Starch contents of the WT and *smk7a* at 12 and 20 DAP and in the mature stage. (**E**) Zein protein contents of the WT and *smk7a* at 12 and 20 DAP and in the mature stage. (**F**) Free IAA contents of the WT and *smk7a* at 12 and 20 DAP. All of the values are the means of the three replicates ± SD. ** denotes statistical significance at *p* < 0.01, and * denotes statistical significance at *p* < 0.05, according to the *t*-test when compared with the values of the WT.

**Figure 4 plants-12-00354-f004:**
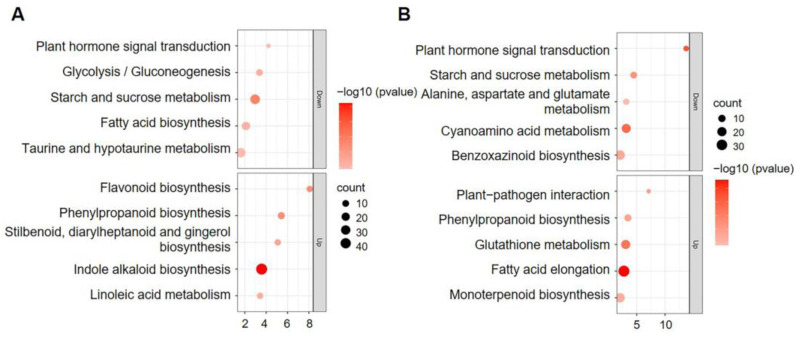
KEGG pathway enrichment analysis of differentially expressed genes between WT and *smk7a* at 12 and 20 DAP. (**A**) Top five KEGG enrichment pathways of the up- and down-regulated DEGs of *smk7a* compared to the WT at 12 DAP. (**B**) Top five KEGG enrichment pathways of up- and down-regulated DEGs of *smk7a* compared to the WT at 20 DAP. The *Y*-axis shows the KEGG pathway. The *X*-axis shows the −log 10 (q-value). The size of each dot represents the number of genes enriched in a particular pathway.

**Figure 5 plants-12-00354-f005:**
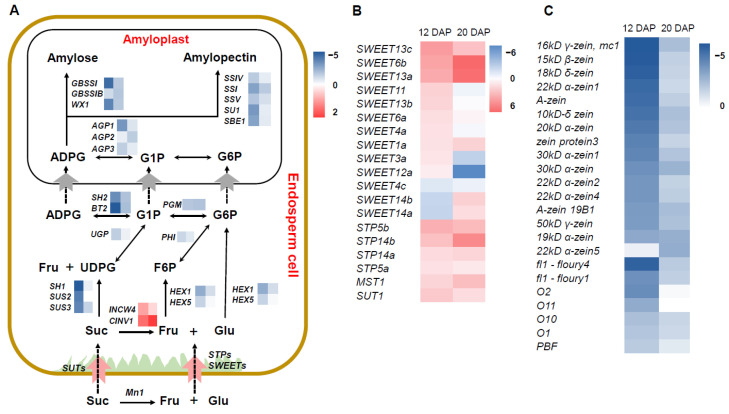
Heat maps of the DEGs that participate in the selected disrupted function or pathway. (**A**) Heat maps of DEGs enriched in starch and sucrose metabolism. The left side color bar represents 12 DAP, the right side color bar represents 20 DAP. (**B**) Heat maps of DEGs with the sugar transport function. (**C**) Heat maps of DEGs with IAA synthesis and signaling functions. The color bar indicates the log2 (fold change) values from small to large.

**Figure 6 plants-12-00354-f006:**
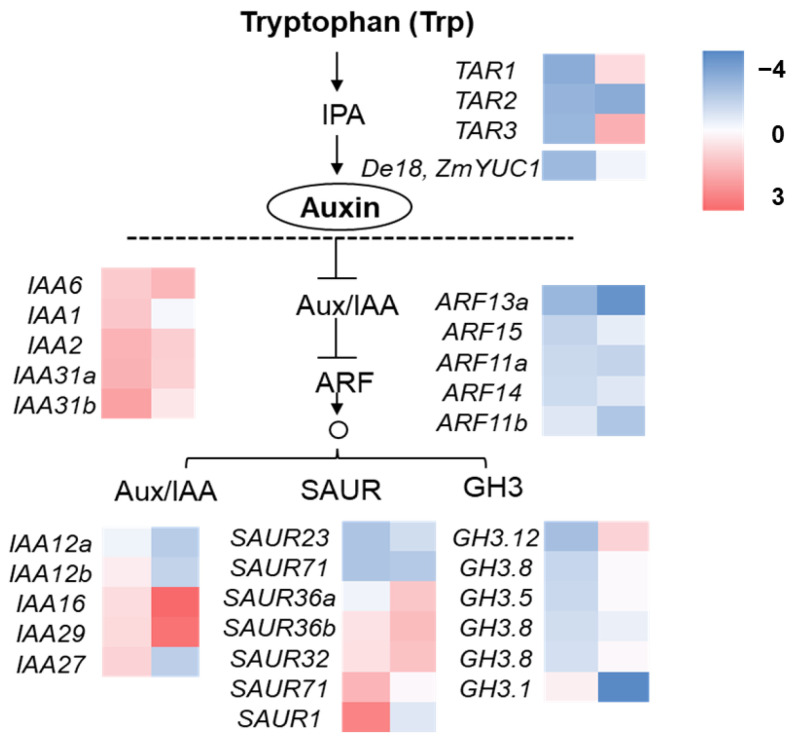
Heat maps of DEGs enriched in IAA synthesis and signaling functions. The color bar indicates the log2 (fold change) values from small to large. The left-side color bar represents 12 DAP, the right-side color bar represents 20 DAP.

**Table 1 plants-12-00354-t001:** Comparison of the endosperm cell area and cell number between the WT and *smk7a* in the same field of vision.

	WT	*smk7a*	Percentage Change	*p* Value
Cell area (μm^2^)	24,616.84	71,077.38	188.7%	9.06 × 10^−6^
Cell number	112	49	−56.3%	2.02 × 10^−3^

## Data Availability

The data presented in this study are available in this article and Appendix A. RNA-seq data have been deposited to the Genome Expression Omnibus (GEO) database, with the accession number of GSE221594.

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
