# Peer review of "Characterization and Transcriptome Analysis of Maize Small-Kernel Mutant smk7a in Different Development Stages"

_plants, 2023, doi:10.3390/plants12020354_

Round 1

Reviewer 1 Report

The paper is of good quality, the transcriptomic analysis is sound and the findings are of interest in the field of maize genomics. However, RNA-seq high throughput sequencing data should be deposited either in the GEO database of in the NCBI's Sequence Read Archive (SRA) for the manuscript to be published. High-quality scientific journals, including Plants, pursue an open science policy of complete availability of data. The Authors must deposit the RNA-seq data in the appropriate databases for the manuscript to be published.

Author Response

Thanks for your comments. We have deposited the RNA-seq data in the GEO database under accession number GSE221594. And we have added this to the “Data Availability Statement” section in the revised manuscript.

Reviewer 2 Report

I can be mistaken because, in the material and methods section, authors do not carefully explain how studied kernels were obtained. If kernels come from selfcrossing the hybrid between the hetrozygous B73/smk7a and Mo17 (or  Chang7-2???), the resulting F2 individuals (kernels) will segregate for many loci besides the mutant locus. Therefore, I do not consider that these kernels are the best materials to do citological, biochemical and transcriptomic studies. If the mutant is in the B73 background, the segregating population for these studies should be obtained by crossing the hetrozygous mutant with B73. 

Author Response

Thanks very much for your comments. We completely agree with your opinions and sorry for the ambiguous description in the “Material and Methods” section. Since the homozygote smk7a cannot grow into a normal plant, the mutants were maintained in heterozygote by continuous self-crossing and phenotype selection. The WT and smk7a kernels used for cytological, biochemical and transcriptome analysis in this study were collected from the same segregating ears of the M8 generation. The genetic background of the WT and smk7a mutant should be homozygotes at more than 99% loci at M8 generation. Therefore, we think our materials are suitable for this study. We also quite agree with that it’s even better to produce segregating population by backcross the heterozygous mutant with B73 for several times. We will pay more attention in our further studies. We have carefully amended the “Materials and Methods” section in revised manuscript, hope that the revision is acceptable.

Round 2

Reviewer 2 Report

Paper is suitable for publication